# LEARNED ISTA WITH ERROR-BASED THRESHOLDING FOR ADAPTIVE SPARSE CODING

## ABSTRACT

The learned iterative shrinkage thresholding algorithm (LISTA) introduces deep unfolding models with learnable thresholds in the shrinkage function for sparse coding. Drawing on some theoretical insights, we advocate an error-based thresholding (EBT) mechanism for LISTA, which leverages a function of the layer-wise reconstruction error to suggest an appropriate threshold value for each observation on each layer. We show that the EBT mechanism well-disentangles the learnable parameters in the shrinkage functions from the reconstruction errors, making them more adaptive to the various observations. With rigorous theoretical analyses, we show that the proposed EBT can lead to faster convergence on the basis of LISTA and its variants, in addition to its higher adaptivity. Extensive experimental results confirm our theoretical analyses and verify the effectiveness of our methods.

## 1 INTRODUCTION

Sparse coding is widely used in many machine learning applications (Xu et al., 2012; Dabov et al., 2007; Yang et al., 2010; Ikehata et al., 2012), and its core problem is to deduce the high-dimensional sparse code from the obtained low-dimensional observation, for example, under the assumption of $y = Ax_s + \varepsilon$, where $y \in \mathbb{R}^m$ is the observation corrupted by the inevitable noise $\varepsilon \in \mathbb{R}^m$, $x_s \in \mathbb{R}^n$ is the sparse code to be estimated, and $A \in \mathbb{R}^{m \times n}$ is an over-complete dictionary matrix. To recover $x_s$ purely from $y$ is called sparse linear inverse problem (SLIP). The main challenge for solving SLIP is its ill-posed nature because of the over-complete modeling, i.e., $m < n$. A possible solution to SLIP can be obtained via solving a LASSO problem using the $l_1$ regularization:

$$\min_x \|y - Ax\|_2 + \lambda \|x\|_1. \tag{1}$$

Possible solutions for Eq. (1) are iterative shrinking thresholding algorithm (ISTA) (Daubechies et al., 2004) and its variants, e.g., fast ISTA (FISTA) (Beck & Teboulle, 2009). Despite their simplicity, these traditional optimization algorithm suffer from slow convergence speed in large scale problems. Therefore, Gregor & LeCun (2010) proposed the learned ISTA (LISTA) which was a deep neural network (DNN) whose architecture followed the iterative process of ISTA. The thresholding mechanism was modified into shrinkage functions in the DNNs together with learnable thresholds. LISTA achieved superior performance in sparse coding, and many theoretical analyses have been proposed to modify LISTA to further improve its performance (Chen et al., 2018; Liu et al., 2019; Zhou et al., 2018; Ablin et al., 2019; Wu et al., 2020).

Yet, LISTA and many other deep networks based on it suffer from two issues. (a) Though the thresholds of the shrinkage functions in LISTA were learnable, their values were shared among all training samples and thus lack adaptability to the variety of training samples and robustness to outliers. According to prior work (Chen et al., 2018; Liu et al., 2019), the thresholds should be proportional to the upper bound of the norm of the current estimation error to guarantee fast convergence in LISTA. However, outliers with drastically higher estimation errors will affect the thresholds more, making the learned thresholds less suitable to other (training) samples. (b) For the same reason, it may also lead to poor generalization to test data with different distribution (or sparsity (Chen et al., 2018)) from the training data. For instance, in practice, we may only be given some synthetic sparse codes but not the real ones for training, and current LISTA models may fail to generalize under such circumstances.

In this paper, we propose an error-based thresholding (EBT) mechanism to address the aforementioned issues of LISTA-based models to improve their performance. Drawing on theoretical insights,

EBT introduces a function of the evolving estimation error to provide each threshold in the shrinkage functions. It has no extra parameter to learn compared with original LISTA-based models yet shows significantly better performance. The main contributions of our paper are listed as follows:

- The EBT mechanism can be readily incorporated into popular sparse coding DNNs (e.g., LISTA (Gregor & LeCun, 2010) and LISTA with support selection (Chen et al., 2018)) to speed up the convergence with no extra parameters.

- We give a rigorous analysis to prove that the estimation error of EBT-LISTA (i.e., a combination of our EBT and LISTA) and EBT-LISTA with support selection (i.e., a combination of our EBT and LISTA with support selection) is theoretically lower than the original LISTA (Gregor & LeCun, 2010) and LISTA with support selection (Chen et al., 2018), respectively. In addition, the introduced parameters in our EBT are well-disentangled from the reconstruction errors and need only to be correlated with the dictionary matrix to ensure convergence. These results guarantee the superiority of our EBT in theory.

- We demonstrate the effectiveness of our EBT in the original LISTA and several of its variants in simulation experiments. We also show that it can be applied to practical applications (e.g., photometric stereo analysis) and achieve superior performance as well.

The organization of this paper is structured as follows. In Section 2, we will review some preliminary knowledge of our study. In Section 3, we will introduce a basic form of our EBT and several of its improved versions. Section 4 provides a theoretical study of the convergence of EBT-LISTA. Experimental results in Section 5 valid the effectiveness of our method in practice. Section 6 summarizes this paper.

## 2    BACKGROUND AND PRELIMINARY KNOWLEDGE

As mentioned in Section 1, ISTA is an iterative algorithm for solving LASSO in Eq. (1). Its update rule is: $x^{(0)} = 0$ and

$$x^{(t+1)} = \mathrm{sh}_{\lambda/\gamma}((I - A^T A/\gamma)x^{(t)} + A^T y/\gamma), \quad \forall t \geq 0, \tag{2}$$

where $\mathrm{sh}_b(x) = \mathrm{sign}(x)(|x| - b)_+$ is a shrinkage function with a threshold $b \geq 0$ and $(\cdot)_+ = \max\{0, \cdot\}$, $\gamma$ is a positive constant scalar greater than or equal to the maximal eigenvalue of the symmetric matrix $A^T A$. LISTA kept the update rule of ISTA but learned parameters via end-to-end training. Its inference process can be formulated as $x^{(0)} = 0$ and

$$x^{(t+1)} = \mathrm{sh}_{b^{(t)}}(W^{(t)}x^{(t)} + U^{(t)}y), \quad t = 0, \ldots, d, \tag{3}$$

where $\Theta = \{W^{(t)}, U^{(t)}, b^{(t)}\}_{t=0,\ldots,d}$ is a set of learnable parameters, and, specifically, $b^{(t)}$ is the layer-wise threshold which is learnable but shared among all samples. LISTA achieved lower reconstruction error between its output and the ground-truth $x_s$ compared with ISTA, and it is proved to convergence linearly (Chen et al., 2018) with $W^{(t)} = I - U^{(t)}A$ holds for any layer $t$. Thus, Eq. (3) can be written as.

$$x^{(t+1)} = \mathrm{sh}_{b^{(t)}}((I - U^{(t)}A)x^{(t)} + U^{(t)}y), \quad t = 0, \ldots, d. \tag{4}$$

Chen et al. (2018) further proposed support selection for LISTA, which introduced $\mathrm{shp}_{(b^{(t)},p)}(x)$ whose elements are defined as

$$(\mathrm{shp}_{(b^{(t)},p)}(x))_i = \begin{cases} \mathrm{sign}(x_i)(|x_i| - b), & \text{if } |x_i| > b, i \notin S_p \\ x_i, & \text{if } |x_i| > b, i \in S_p \ , \\ 0, & \text{otherwise} \end{cases} \tag{5}$$

to substitute the original shrinking function $\mathrm{sh}_{b^{(t)}}(x)$, where $S_p$ is the set of the index of the largest $p\%$ elements (in absolute value) in vector $x$. Formally, the update rule of LISTA with support selection is formulated as $x^{(0)} = 0$ and

$$x^{(t+1)} = \mathrm{shp}_{(b^{(t)},p^{(t)})}((I - UA)x^{(t)} + U^{(t)}y), \quad t = 0, \ldots, d, \tag{6}$$

where $p^{(t)}$ is a hyper-parameter and it increases from early layers to later layers. LISTA with support selection can achieve faster convergence compared with LISTA (Chen et al., 2018).

Theoretical studies (Chen et al., 2018; Liu et al., 2019) also demonstrate that the threshold of LISTA and its variants should satisfy the equality

$$b^{(t)} \leftarrow \mu(A) \sup_{x_s \in \mathcal{S}} \|x^{(t)} - x_s\|_p \tag{7}$$

to ensure fast convergence in the noiseless case (i.e., $\varepsilon = \mathbf{0}$), where $\mathcal{S}$ is the training set and $\mu(A)$ is the general mutual coherence coefficient of the dictionary matrix $A$. Note that $\mu(A)$ is a crucial term in this paper, here we formally give its definition together with the definition of $\mathcal{W}(A)$ as follows.

**Definition 1** *For $A \in \mathbb{R}^{m \times n}$, its generalized coherence coefficient is defined as $\mu(A) = \inf_{W \in \mathbb{R}^{n \times m}, W_{i,:} A_{:,i} = 1} \max_{i \neq j} |(W_{i,:} A_{:,j})|$, and we say $W \in \mathcal{W}(A)$ if $\max_{i \neq j} (W_{i,:} A_{:,j}) = \mu(A)$.*

## 3 METHODS

In LISTA and its variants, the threshold $b^{(t)}$ is commonly treated as a learnable parameter. As demonstrated in Eq. (7), $b^{(t)}$ should be proportional to the upper bound of the estimation error of the $t$-th layer in the noiseless case to ensure fast convergence. Thus, some outliers or extreme training samples largely influence the value of $b^{(t)}$, making the obtained threshold not fit the majority of the data. To be specific, we know that the suggested value of $b^{(t)}$ is $b^{(t)} = \mu(A) \sup_{i=0,1,\dots,n} \|x_i^{(t)} - x_{s_i}\|_p$ for $\|x\|_1$ a training set $\{x_{s_i}\}_{i=0,1,\dots,n}$, and normal training of LISTA leads to it in theory (Chen et al., 2018). Yet, if a new training sample $x_{s_{n+1}}$ with higher reconstruction error is introduced, the expected $b^{(t)}$ shall change to $\mu(A) \|x_{n+1}^{(t)} - x_{s_{n+1}}\|_p$, which is probably undesirable for the other samples. Similar problems occur if there exists a large variety in the value of reconstruction errors.

In order to solve this problem, we propose to disentangle the reconstruction error term from the learnable part of the threshold and introduce adaptive thresholds for LISTA and related networks. We attempt to rewrite the threshold at the $t$-th layer as something like

$$b^{(t)} = \rho^{(t)} \|x^{(t)} - x_s\|_p, \tag{8}$$

where $\rho^{(t)}$ is a layer-specific learnable parameter. However, the ground-truth $x_s$ is actually unknown for the inference process in SLIP. Therefore, we need to find an alternative formulation. Notice that in the noiseless case, it holds that $Ax^{(t)} - y = A(x^{(t)} - x_s)$, thus we further rewrite Eq. (8) into

$$b^{(t)} = \rho^{(t)} \|Q(Ax^{(t)} - y)\|_p, \tag{9}$$

where $Q \in \mathbb{R}^{n \times m}$ is a compensation matrix introduced to let Eq. (9) approximate Eq. (8) better, i.e., a matrix that makes $QA$ approaches the identity matrix is more desired. However, note that although $QA$ is a low-rank matrix and can never be an identity matrix, we can encourage its diagonal elements to be 1 and the non-diagonal elements to be nearly zero. This can be directly achieved by letting $Q \in \mathcal{W}(A)$, where $\mathcal{W}(A)$ is defined in Definition 1. According to some prior works (Liu et al., 2019; Wu et al., 2020; Chen et al., 2018), we also know that $U^{(t)} \in \mathcal{W}(A)$ to guarantee linear convergence, thus $U^{(t)}$ can probably be a reasonable option for the matrix $Q$ in our method, making it layer-specific as well. Therefore, our EBT-LISTA is formulated as $x^{(0)} = 0$ and

$$\begin{aligned} x^{(t+1)} &= \mathrm{sh}_{b^{(t)}}((I - U^{(t)}A)x^{(t)} + U^{(t)}y), \quad t = 0, \dots, d, \\ b^{(t)} &= \rho^{(t)} \|U^{(t)}(Ax^{(t)} - y)\|_p. \end{aligned} \tag{10}$$

Note that only $\rho^{(t)}$ and $U^{(t)}$ are learnable parameters in the above formulation, thus our EBT-LISTA actually introduces no extra parameters compared with the original LISTA. The architecture of LISTA and EBT-LISTA are shown in Figure 1. We can also apply our EBT mechanism on LISTA with support selection (Chen et al., 2018). It is straightforward to keep the support set selection operation and replace the fixed threshold with our EBT, and such a combination can be similarly formulated as $x^{(0)} = 0$ and

$$\begin{aligned} x^{(t+1)} &= \mathrm{shp}_{(b^{(t)}, p^{(t)})}((I - U^{(t)}A)x^{(t)} + U^{(t)}y), \quad t = 0, \dots, d, \\ b^{(t)} &= \rho^{(t)} \|U^{(t)}(Ax^{(t)} - y)\|_p. \end{aligned} \tag{11}$$

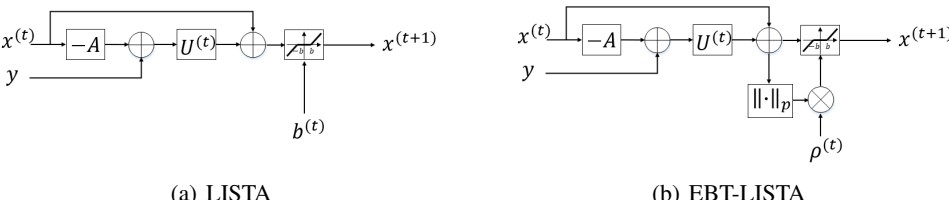

(a) LISTA (b) EBT-LISTA

Figure 1: The $t$-th layer of LISTA and EBT-LISTA.

Our former analysis is based on the noiseless case. For noise case, there is $A(x^{(t)} - x_s) = Ax^{(t)} - y + \varepsilon$. Since the noise is generally unknown in practical problems, we may add an extra learnable parameter on the threshold to compensate the noise, i.e.,

$$b^{(t)} = \rho^{(t)} \|U^{(t)}(Ax^{(t)} - y)\|_p + \alpha^{(t)}, \tag{12}$$

where $\alpha^{(t)}$ is the learnable parameter for the observation noise.

## 4 THEORETICAL ANALYSIS

In this section, we provide convergence analyses for LISTA and LISTA with support selection. We focus on the noiseless case and the main results are obtained under a mild assumption of the ground-truth sparse code. To be more specific, we assume that the ground-truth sparse vector $x_s$ is sampled from the distribution $\gamma(B, s)$, i.e., the number of its nonzero elements follows a uniform distribution $U(0, s)$ and the magnitude of its nonzero elements follows a arbitrary distribution in $[-B, B]$. Compared with the assumptions made in some recent work (Chen et al., 2018; Liu et al., 2019; Wu et al., 2020), i.e., $\mathcal{X}(B, s) = \{x_s | \|x_s\|_0 \leq s, \|x_s\|_\infty \leq B\}$, our assumption here provides a more detailed yet also stricter description of the distribution of $x_s$, especially for the sparsity of $x_s$. In fact, it can be easily derived that our assumption can also be rewritten in the form of some prior assumptions. In particular, for $\forall x_s \in \gamma(B, s)$, there exist $x_s \in \mathcal{X}(B, s) := \{x_s | \|x_s\|_0 \leq s, \|x_s\|_\infty \leq B\}$ (Wu et al., 2020). We will mention the condition that $s$ is sufficiently small for error-based thresholding, which means $\mu(A)s \ll 1$ specifically.

### 4.1 ERROR-BASED THRESHOLDING ON LISTA

Let us first discuss the convergence of LISTA and how our EBT improves LISTA in accelerating convergence. Proof of all our theoretical results can be found in Appendix A.2. To get started, we would like to recall the theoretical guarantee of the convergence of LISTA, which was given in the work of Chen et al. (2018)'s.

**Lemma 1** *(Chen et al., 2018). For LISTA formulated in Eq. (4), if $x_s$ is sampled from $\gamma(B, s)$ and $s$ is small such that $\mu(A)(2s-1) < 1$, with $U^{(t)} \in \mathcal{W}(A)$, assume that $b^{(t)} = \mu(A)\sup_{x_s} \|x^{(t)} - x_s\|_1$ is achieved to guarantee the "no false positive" property (i.e. $\mathrm{supp}(x^{(t)}) \subset \mathrm{supp}(x_s)$), then the estimation $x^{(t)}$ at the $t$-th layer of LISTA satisfies*

$$\|x^{(t)} - x_s\|_2 \leq sB \exp(c_0 t),$$

*where $c_0 = \log((2s - 1)\mu(A)) < 0$.*

The above lemma shows that the reconstruction error of LISTA decreases at a rate of $c_0$ and the reconstruction error bound is also related to $s$ and $B$. With Lemma 1, we further show in the following theorem that the convergence of EBT-LISTA is similarly bounded from above.

**Theorem 1 (Convergence of EBT-LISTA)** *For EBT-LISTA formulated in Eq. (10), for $p = 1$, if $x_s$ is sampled from $\gamma(B, s)$ and $s$ is sufficiently small, with $U^{(t)} \in \mathcal{W}(A)$, assume $\rho^{(t)} = \frac{\mu(A)}{1 - \mu(A)s}$ is achieved to guarantee the "no false positive" property, then the estimation $x^{(t)}$ at the $t$-th layer satisfies*

$$\|x^{(t)} - x_s\|_2 \leq q_0 \exp(c_1 t),$$

*where $q_0 < sB$ and $c_1 < c_0$ hold with the probability of $1 - \mu(A)s$.*

From Theorem 1 we know that EBT-LISTA converges similarly with a rate of $c_1$, which is probably faster than that of the original LISTA. In addition, we show that its reconstruction error at each layer with an index $t$ is also probably lower than that of the original LISTA, with $q_0 < sB$. Since $\mu(A)$ is generally small in practical and $s$ is assumed to be sufficiently small such that $\mu(A)s \ll 1$, the probability of achieving the superiority is very high in theory. We will further show in experiments that even if $s$ is not that small (i.e., the sparsity is not that high), our EBT can still achieve favorable improvement on the basis of LISTA. Moreover, unlike the desired threshold in the original LISTA (i.e., $\mu(A) \sup_{x_s} \|x^{(t)} - x_s\|_1$) which depends on specific training samples, the desired threshold in EBT-LISTA is disentangled with the reconstruction error.

### 4.2 ERROR-BASED THRESHOLD ON LISTA WITH SUPPORT SELECTION

Effective prior work from Chen et al. (2018) proposed to improve the performance of LISTA with the operation of support set selection, and it helps to achieve lower error bound when compared with the original LISTA. We first give a detailed discussion on the convergence of LISTA combined with support selection before we delve deeper into the theoretical study of how combining our EBT with them further improves the performance.

**Lemma 2 (Convergence of LISTA with support selection)** *For LISTA with support selection as formulated in Eq. (6), if $x_s$ is sampled from $\gamma(B, s)$ and $s$ is sufficiently small, with $U^{(t)} \in \mathcal{W}(A)$, assume that $b^{(t)} = \mu(A) \sup_{x_s} \|x^{(t)} - x_s\|_1$ is achieved and $p^{(t)}$ is sufficiently large, there actually exist two convergence phases.*

*In the first phase, i.e., $t \leq t_0$, the $t$-th layer estimation $x^{(t)}$ satisfies*

$$\|x^{(t)} - x_s\|_2 \leq sB \exp(c_2 t),$$

*where $c_0 \leq \log((2s-1)\mu(A))$. In the second phase, i.e., $t > t_0$, the estimation $x^{(t)}$ satisfies*

$$\|x^{(t)} - x_s\|_2 \leq C\|x^{(t-1)} - x_s\|_2,$$

*where $C \leq s\mu(A)$.*

Lemma 2 shows that when powered with support selection, LISTA shows two different convergence phases. The earlier phase is generally slower and the later phase is faster. On the basis of the theoretical results in Lemma 2, we further show in the following theorem that the convergence of EBT-LISTA with support selection is similarly bounded and also has two phases.

**Theorem 2 (Convergence of EBT-LISTA with support selection)** *For EBT-LISTA with support selection and $p = 1$, if $x_s$ is sampled from $\gamma(B, s)$ and $s$ is sufficiently small, with $U^{(t)} \in \mathcal{W}(A)$, assume $\rho^{(t)} = \frac{\mu(A)}{1-\mu(A)s}$ is achieved and $p^{(t)}$ is sufficiently large, there exist two convergence phases.*

*In the first phase, i.e., $t \leq t_1$, the $t$-th layer estimation $x^{(t)}$ satisfies*

$$\|x^{(t)} - x_s\|_2 \leq q_1 \exp(c_3 t),$$

*where $c_3 < c_2$, $q_1 < sB$ and $t_1 < t_0$ hold with a probability of $1 - \mu(A)s$. In the second phase, i.e., $t > t_1$, the estimation $x^{(t)}$ satisfies*

$$\|x^{(t)} - x_s\|_2 \leq C\|x^{(t-1)} - x_s\|_2,$$

*where $C \leq s\mu(A)$.*

The above theorem shows that further incorporated with EBT, the model shall also show two different phases of convergences. In addition, it processes the same rate of convergence in the second phase, comparing with the results of the original LISTA with support selection in Lemma 2, while in the first phase, our EBT leads to faster convergence and faster enter the second phase, which shows the effectiveness of our EBT in LISTA with support selection in theory.

## 5 EXPERIMENTS

We conduct extensive experiments on both synthetic data and real data to validate our theorem and testify the effectiveness of our methods. The network architectures and training strategies in our experiments most follow those of prior works (Chen et al., 2018; Wu et al., 2020). To be more specific,

all the compared networks have $d = 16$ layers and their learnable parameters $W^{(t)}, U^{(t)}, \rho^{(t)}$ (or $b^{(t)}$ without out EBT) are not shared between layers. The training batch size is 64, and we use the popular Adam optimizer (Kingma & Ba, 2014) for training with its default hyper-parameters $\beta_1 = 0.9$, $\beta_2 = 0.999$. The training is performed from layer to layer in a progressive way, i.e., if the validation loss of the current layer does not decrease for 4000 iterations, the training will move to the next layer. When training each layer, the learning rate is first initialized to be 0.0005 and it will then decrease to 0.0001 and finally decrease to 0.00001, if the validation loss does not decrease for 4000 iterations. Specifically, in our proposed methods, we impose the constraint between $W^{(t)}$ and $U^{(t)}$ and make sure it holds that $W^{(t)} = I - U^{(t)}A, \forall t$, i.e., the coupled constraints are introduced for all the evaluated models. For models empowered with support selection, we will append "SS" to their name for clarity, i.e., LISTA with support selection is renamed "LISTA-SS" in the following discussions. The value of $\rho^{(t)}$ is initialized to be 0.02.

### 5.1 SIMULATION EXPERIMENTS

**Basic settings.** In simulation experiments, we set $m = 250, n = 500$, and we generate the dictionary matrix $A$ by using the standard Gaussian distribution. The indices of the non-zero entries in $x_s$ are determined by a Bernoulli distribution letting its sparsity (i.e., the probability of any of its entry be zero) be $p_b$, while the magnitudes of the non-zero entries are also sampled from the standard Gaussian distribution. The noise $\varepsilon$ is sampled from a Gaussian distribution where the standard deviation is determined by the noise level. With $y = Ax_s + \varepsilon$, we can randomly synthesize in-stream $x_s$ and get a corresponding set of observations for training. Similarly, we also synthesize two sets for validation and test, respectively, each contains 1000 samples. The sparse coding performance of different models is evaluated by the normalized mean squared error (NMSE) in decibels (dB):

$$\text{NMSE}(x, x_s) = 10 \log_{10} \left( \frac{\|x - x_s\|_2^2}{\|x\|_2^2} \right). \tag{13}$$

**Disentanglement.** First we would like to compare the learned parameters (i.e., $b^{(t)}$ and $p^{(t)}$) for thresholds in both LISTA and our EBT-LISTA. Figure 2(a) shows how the learned parameters (in a logarithmic coordinate) vary with the index of layers in LISTA and EBT-LISTA. Note that the mean values are removed to align the range of the parameters of different models on the same y-axis. It can be seen that the obtained values for the parameter in EBT-LISTA do not change much from lower layers to higher layers, while the reconstruction errors in fact decrease. By contrast, the obtained threshold values in LISTA vary a lot across layers. Such results imply that the optimal thresholds in EBT-LISTA are indeed independent to (or say disentangled from) the reconstruction error, which confirms the theoretical result in Theorem 1. Similar observation can also be made on LISTA-SS (i.e., LISTA with support selection) and our EBT-LISTA-SS (i.e., EBT-LISTA with support selection), as shown in Figure 2(b).

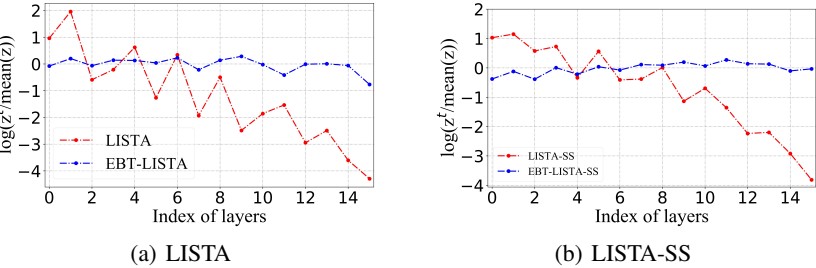

|          (a) LISTA          |          (b) LISTA-SS          |

Figure 2: Disentanglement of the reconstruction error and learnable parameters in our EBT. $z^{(t)}$ here indicates $\rho^{(t)}$ and $b^{(t)}$ for networks with or without EBT, respectively.

We analyze the obtained threshold values in EBT-LISTA and EBT-LISTA-SS, i.e., $b^{(t)} = \rho^{(t)} \|U^{(t)}(Ax^{(t)} - y)\|_p$ (with $p = 2$), and we compare them with the thresholds values obtained in LISTA and LISTA-SS. Note that the threshold values in our EBT-based models differ from sample to sample, we show the results in Figure 3. It can be seen that the learned thresholds in our EBT-based methods and the original LISTA and LISTA-SS are similar, which indicates that the introduced EBT mechanism does not modify the training dynamics of the original methods, and our EBT works by disentangling the reconstruction error and learnable parameters.

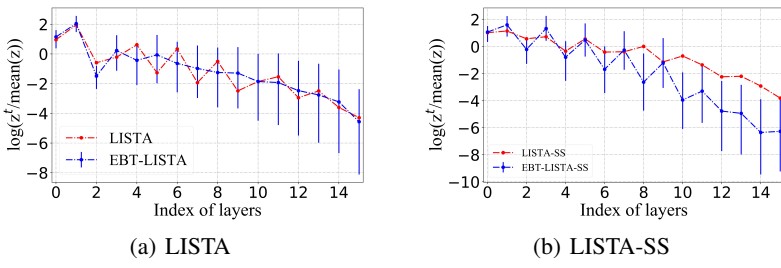

(a) LISTA

(b) LISTA-SS

Figure 3: Thresholds obtained from different methods across layers.

.

**Validation of Theorem 2.** Figure 4(a) shows how the NMSE of EBT-LISTA-SS varies with the index of layers. Besides the $l_1$ norm (i.e., $b^{(t)} = \rho^{(t)}\|U^{(t)}(Ax^{(t)} - y)\|_1$) concerned in the theorem, we also test EBT-LISTA-SS with the $l_2$ norm (i.e., letting $b^{(t)} = \rho^{(t)}\|U^{(t)}(Ax^{(t)} - y)\|_2$). It can be seen that, with both the $l_1$ and $l_2$ norms, EBT-LISTA-SS leads to consistently faster convergence than LISTA. Also, it is clear that there exist two convergence phases for EBT-LISTA-SS and LISTA-SS, and the later phase is indeed faster. With faster convergence, EBT-LISTA-SS finally achieves superior performance. The experiment is performed in the noiseless case with $p_b = 0.95$. Similar observations can be made on the basis of other variants of LISTA (e.g., ALISTA, see Figure 4(b)).

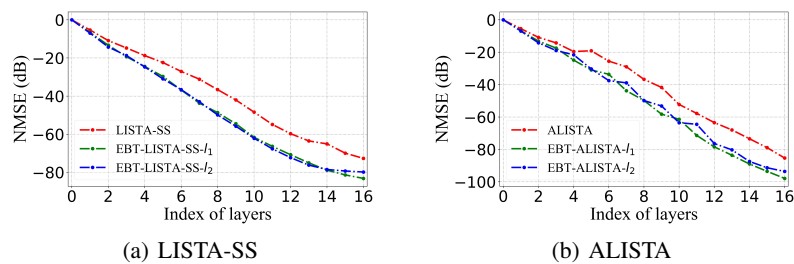

(a) LISTA-SS

(b) ALISTA

Figure 4: Validation of Theorem 2: there exist two convergence phases and our EBT accelerates the convergence of LISTA-SS, in particular in the first phase.

.

**Adaptivity to unknown sparsity.** As have been mentioned, in some practical scenarios, there may exist a gap between the training and test data distribution, or we may not know the distribution of real test data and will have to train on synthesized data based on the guess of the test distribution. Under such circumstances, it is of importance to consider the adaptivity/generalization of the sparse coding model (trained on a specific data distribution or with a specific sparsity) to test data sampled from different distributions with different sparsity. We conduct an experiment to test in such a scenario, in which we let the test sparsity be different from the training sparsity. Figure 5 shows the results in three different settings. The black curves represent the optimal model when LISTA is trained on exactly the same sparsity as that of the test data. It can be seen that our EBT has huge advantages in such a practical scenario where the adaptivity to un-trained sparsity is required, and the performance gain of LISTA is larger when the distribution shift between training and test is larger (cf. purple line and yellow line in Figure 5(a) and 5(b)).

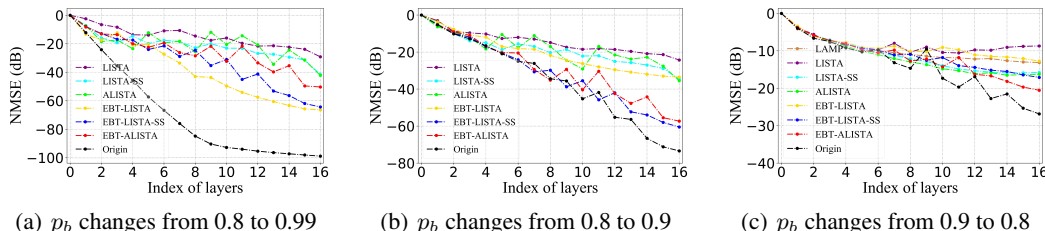

(a) $p_b$ changes from 0.8 to 0.99    (b) $p_b$ changes from 0.8 to 0.9    (c) $p_b$ changes from 0.9 to 0.8

Figure 5: NMSE of different methods when the test sparsity is different from the training sparsity. We use "Origin" to indicate the optimal scenario where the LISTA models are trained on exactly the same sparsity as that of the test data.

**Comparison with competitors.** Here we compare EBT-LISTA-SS, EBT-LISTA, and EBT-ALISTA with other methods comprehensively. In addition to LISTA-SS, LISTA, and ALISTA, we also compare with learned AMP (LAMP) (Borgerding et al., 2017) here. The performance of different net-

works under different noise levels are shown in Figure 6. It can be seen that when combined with LISTA and its variants, our EBT achieves better or similar performance. Figure 6(a) demonstrates that the combination of ALISTA and EBT performs the best in the noiseless case (i.e., SNR=∞), yet it is inferior to the other networks when noise presents. The figures also show that the performance of our EBT is more promising in the noiseless or low noise cases (i.e., SNR=∞ and SNR=40dB), while in a very noisy scenarios it provides little help.

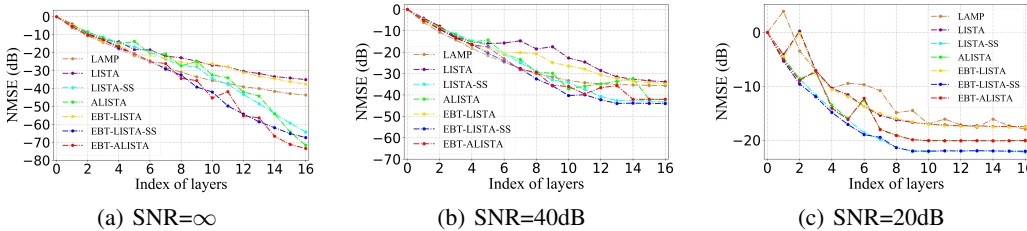

(a) SNR=∞          (b) SNR=40dB          (c) SNR=20dB

Figure 6: NMSE of different sparse coding methods under different noise levels. It can be seen that our EBT performs favorably well under SNR=∞ and SNR=40dB.

We further test different networks under different sparsity and different condition numbers. Note that for the methods with support selection (i.e., EBT-ALISTA, EBT-LISTA-SS, ALISTA, and LISTA-SS), $p$ and $p_{max}$ are set as 0.6 and 6.5 when $p_b = 0.95$, and are set as 1.2 and 13.0 when $p_b = 0.9$. Figure 7 demonstrates some of the results in different settings, while more results can be found in Appendix A.1. In all settings, we can see that our EBT leads to significantly faster convergence. In addition, the superiority of our EBT-based models is more significant with a larger $p_b$ for which the assumption of a sufficiently small $s$ is more likely to hold (comparing Figure 7(a), 9(a), and 9(b)). We also tried training with $p_b = 0.99$, yet we found that some classical models failed to converge in such a setting so the results are now shown here.

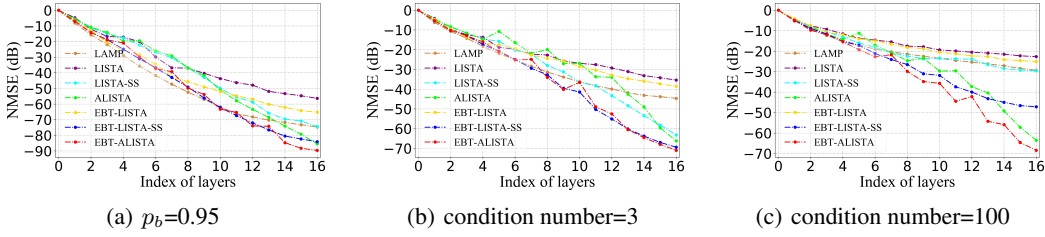

(a) $p_b$=0.95          (b) condition number=3          (c) condition number=100

Figure 7: NMSE of different sparse coding methods in different settings where different sparsity and different condition numbers are considered. When we vary the condition numbers, we fix $p_b$=0.9.

## 5.2 PHOTOMETRIC STEREO ANALYSIS

We also consider a practical sparse coding task: photometric stereo analysis (Ikehata et al., 2012). The task solves the problem of estimating the normal direction of a Lambertian surface, given $q$ observations under different light directions. It can be formulated as

$$o = \rho Ln + e, \tag{14}$$

where $o \in \mathbb{R}^q$ is the observation, $n \in \mathbb{R}^3$ represents the normal direction of the Lambertian surface which is to be estimated, $L \in \mathbb{R}^{q \times 3}$ represents the normalized light directions, $e$ is the noise vector, and $\rho$ is the diffuse albedo scalar. Although the normal vector $n$ is unconstrained in Eq. (14), the noise vector $e$ is found to be generally sparse (Wu et al., 2010; Ikehata et al., 2012). Therefore, we may estimate the noise $e$ first. We introduce the orthogonal complement of $L$, denoted by $L^\dagger$, to rewrite Eq. (14) as

$$L^\dagger o = \rho L^\dagger Ln + L^\dagger e = L^\dagger e. \tag{15}$$

On the basis of the above equation, the estimation of $e$ is basically a sparse coding problem in the noiseless case, where $L^\dagger$ is the dictionary matrix $A$, $e$ is the sparse code $x_s$ to be estimated in the

reformulated problem, and $L^\dagger o$ is the observation $y$. Once we have achieved a reasonable estimation of $e$, we can further obtain $n$ by using the equation $n = L^\dagger(o - e)$.

In this experiment, we mainly follow the settings in Xin et al. 2016's and Wu et al. 2020's work. We use the same bunny picture for evaluation and $L$ is also randomly selected from the hemispherical surface. We set the number of observations $q$ to be 15, 25, and 35 and let the training sparsity of $e$ be $p_t = 0.8$. The final performance is evaluated by calculating the average angle between the estimated normal vector and the ground-truth normal vector (in degree). Since the distribution of the noise is generally unknown in practice, the adaptivity is of great importance for this task. We use two test settings for evaluating different models, in which the sparsity of the noise in test data (i.e., $p_e$) is set as 0.8 and 0.9. We compare EBT-LISTA-SS, LISTA-SS, and two conventional methods including using the least squares (codenamed: $l_s$) and least 1-norm (codenamed: $l_1$) in Table 1. Also, we build the reconstruction 3D error maps for LISTA-SS and EBT-LISTA-SS, shown in Figure 8 in the appendix. The results show that EBT-LISTA-SS outperforms all the competitors in all the concerned settings, note that the advantage is remarkable when $p_e = 0.9$, which means our EBT-based network has better adaptivity and can be more effective in this practical tasks.

Table 1: Mean error (in degree) with different number of observations and different test sparsity

| $p_e$ | $q$ | $l_s$ | $l_1$ | LISTA-SS | EBT-LISTA-SS |
|---|---|---|---|---|---|
| | 15 | 3.41 | 0.678 | $5.50 \times 10^{-2}$ | $4.09 \times 10^{-2}$ |
| 0.8 | 25 | 3.05 | 0.408 | $7.48 \times 10^{-3}$ | $3.17 \times 10^{-3}$ |
| | 35 | 2.78 | 0.336 | $1.89 \times 10^{-3}$ | $5.95 \times 10^{-4}$ |
| | 15 | 1.94 | 0.232 | $6.67 \times 10^{-3}$ | $2.57 \times 10^{-3}$ |
| 0.9 | 25 | 0.145 | 2.03 | $1.33 \times 10^{-3}$ | $1.64 \times 10^{-4}$ |
| | 35 | 1.61 | 0.088 | $2.93 \times 10^{-4}$ | $4.91 \times 10^{-5}$ |

## 6 CONCLUSION

In this paper, we have studied the thresholds in the shrinkage functions of LISTA. We have proposed a novel EBT mechanism that well-disentangles the learnable parameter in the shrinkage function on each layer of LISTA from its layer-wise reconstruction error. We have proved theoretically that, in combination with LISTA and its variants, our EBT mechanism leads to faster convergence and achieves superior final sparse coding performance. Also, we have shown that the EBT mechanisms endow deep unfolding models higher adaptivity to different observations with a variety of sparsity. Our experiments on both synesthetic data and real data have testified the effectiveness of our EBT, especially when the distribution of the test data is different from that of the train data. We hope to extend our EBT mechanism to more complex tasks in future work.

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

# A   APPENDIX

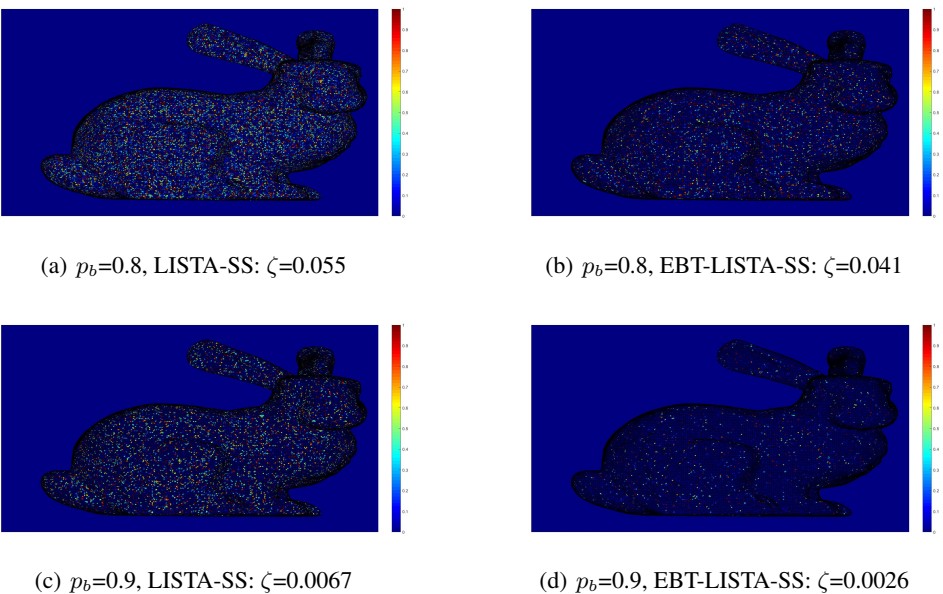

(a) $p_b$=0.8, LISTA-SS: $\zeta$=0.055           (b) $p_b$=0.8, EBT-LISTA-SS: $\zeta$=0.041

(c) $p_b$=0.9, LISTA-SS: $\zeta$=0.0067           (d) $p_b$=0.9, EBT-LISTA-SS: $\zeta$=0.0026

Figure 8: Reconstruction 3D error maps of different methods in different settings. $\zeta$ here is the mean estimation error in degree. Note that the maximal error is 0.1 and 0.03 in theory when $p_b = 0.8$ and $p_b = 0.9$, respectively.

## A.1   ADDITIONAL SIMULATION RESULTS

**Comparison with competitors.** More experiment results than those in Figure 7 are given here. Figure 9 shows the results of our methods with EBT (i.e. EBT-LISTA, EBT-LISTA-SS, and EBT-ALISTA) and other competitors in settings including $p_b$=0.9, $p_b$=0.8 (i.e. the sparsity is 0.9 and 0.8), and when the condition number is set as 3 (with $p_b = 0.9$). For those concerned methods with support selection (EBT-ALISTA, EBT-LISTA-CPSS, ALISTA, and LISTA-CPSS), $p$ and $p_{max}$ are set as 1.2(1.5) and 13(16.25) when $p_b = 0.9(0.8)$. From Figure 9, we can find that our EBT leads to better performance as shown in the results in the main paper.

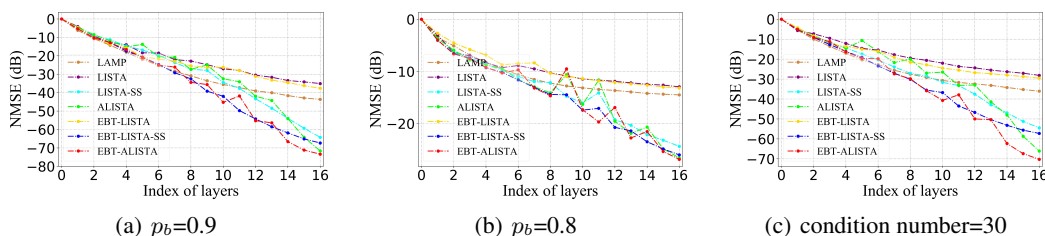

(a) $p_b$=0.9           (b) $p_b$=0.8           (c) condition number=30

Figure 9: NMSE of different sparse coding methods in different settings where different sparsity and different condition numbers are considered.

**Random sparsity.** We also consider the scenario where the sparsity of the data follows a certain distribution. We test with two distributions of sparsity (i.e., $p_b$): $p_b \sim U(0.9, 1)$ (uniform distribution) and $p_b \sim N(0.95, 0.025) \in [0.9, 1]$ (truncated normal distribution). The comparison results in such settings are shown in Figure 10. Our EBT lead to huge advantages in all the methods and setting, indicating that the conclusion in Theorem 1 can be extended to broader distributions of the data sparsity.

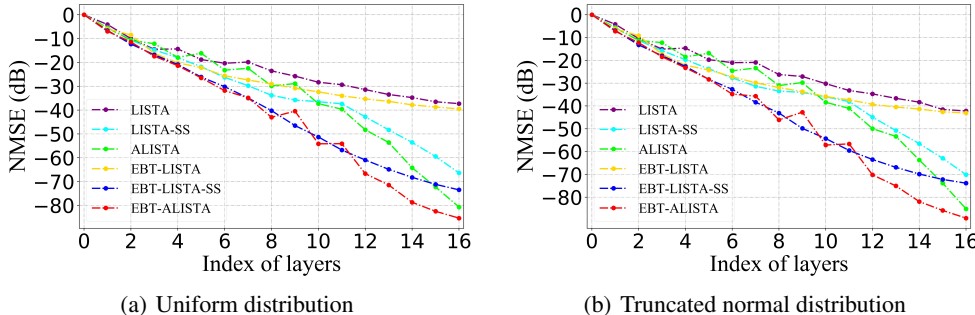

(a) Uniform distribution         (b) Truncated normal distribution

Figure 10: NMSE of different sparse coding methods when the sparsity of the data follows a certain distribution.

**EBT mechanism on (F)ISTA.** We further conduct our proposed EBT mechanism on standard ISTA and FISTA. Note that ISTA and FISTA are nonlineared convergence. We set the scalar $\gamma$ (in Eq. 2) as a constant and the regularization coefficients $\lambda$ (in Eq. 1 and Eq. 2) as $0.1$ and $0.2$. In our proposed methods, we set $\lambda\|Ax^{(t)} - y\|_1/\gamma$ as the thresholds, comparaed with $\lambda/\gamma$ in (F)ISTA. From the experiment results shown in Figure 11, we can find that our EBT mechanism leads to faster convergence. However the fast convergence have negative influence on the final performances. That is because the convergence speed of ISTA and FISTA are sub-lineared, while our EBT mechanism is proposed for linear convergence methods. Therefore, EBT-(F)ISTA might convergence too fast to reach a better performance.

## A.2 PROOF OF THEOREMS

We first give some important notations before we delve into the proofs. The support set is defined as the index set of non-zero values of $x$ and it is written as $\mathrm{supp}(x)$. We define $\mathcal{S}$ as the support set of the vector $x_s$, and we further let $|\mathcal{S}|$ denote the element number of $\mathcal{S}$. We denote $(x)_i$ as i-th element of the vector $x$ and denote $(M)_{ij}$ as the element from the i-th row and j-th column of the matrix $M$.

### A.2.1 PROOF OF THEOREM 1

Assume $i \notin \mathcal{S}$, i.e., $(x_s)_i = 0$. If $(x^{(t+1)})_i \neq 0$ , note that $y = Ax_s$, there is,

$$
\begin{aligned}
b^{(t)} = \rho^{(t)}\|U^{(t)}(Ax^{(t)} - y)\|_1 &< |(x^{(t+1)})_i| \\
&< |[(I - U^{(t)}A)x^{(t)} + U^{(t)}Ax_s]_i| \\
&= |[(I - U^{(t)}A)(x^{(t)} - x_s) + x_s]_i| \\
&\leq |[(I - U^{(t)}A)(x^{(t)} - x_s)]_i| + |(x_s)_i| \\
&= |[(I - U^{(t)}A)(x^{(t)} - x_s)]_i| \\
&= |\sum_j (I - U^{(t)}A)_{ij}(x^{(t)} - x_s)_j| \qquad (16)\\
&\leq \sum_j |(I - U^{(t)}A)_{ij}(x^{(t)} - x_s)_j| \\
&\leq \sum_j \mu(A)|(x^{(t)} - x_s)_j| \\
&\leq \mu(A)\|x^{(t)} - x_s\|_1.
\end{aligned}
$$

From above, we also have $[(I - U^{(t)}A)(x^{(t)} - x_s)]_i \leq \mu(A)\|x^{(t)} - x_s\|_1$, therefore we have $\|(I - U^{(t)}A)(x^{(t)} - x_s)\|_1 \leq |S|\mu(A)\|x^{(t)} - x_s\|_1$. Since $\|U^{(t)}A(x^{(t)} - x_s)\|_1 = \|(x^{(t)} - x_s) - (I - U^{(t)}A)(x^{(t)} - x_s)\|_1$, we have

$$
(1 - |S|\mu(A))\|x^{(t)} - x_s\|_1 \leq \|U^{(t)}A(x^{(t)} - x_s)\|_1 \leq (1 + |S|\mu(A))\|x^{(t)} - x_s\|_1. \qquad (17)
$$

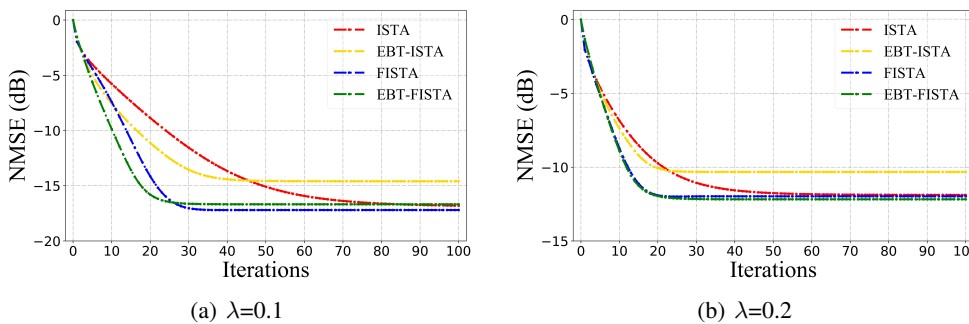

(a) $\lambda$=0.1          (b) $\lambda$=0.2

Figure 11: NMSE of different sparse coding methods where different regularization coefficients $\lambda$ are considered.

Since $\rho^{(t)} = \frac{\mu(A)}{1-\mu(A)s} \geq \frac{\mu(A)}{1-\mu(A)|S|}$, there is

$$\rho^{(t)}\|U^{(t)}A(x^{(t)} - x_s)\|_1 \geq \mu(A)\|x^{(t)} - x_s\|_1. \tag{18}$$

Eq. (16) and (18) are conflicted, which means $(x^{(t+1)})_i = 0$ if $(x_s)_i = 0$ (i.e. $\operatorname{supp}(x^{(t+1)}) \subset \mathcal{S})$, which means our EBT-LISTA is also "no false positive".

From Eq. (10), we have

$$
\begin{aligned}
x^{(t+1)} - x_s &= \mathrm{sh}_{b^{(t)}}((I - U^{(t)}A)x^{(t)} + U^{(t)}y) - x_s \\
&= (I - U^{(t)}A)x^{(t)} + U^{(t)}Ax_s - x_s - b^{(t)} \odot h(x^{(t+1)}) \\
&= (I - U^{(t)}A)(x^{(t)} - x_s) - b^{(t)} \odot h(x^{(t+1)}),
\end{aligned}
\tag{19}
$$

where $h(x) = 1$ if $x > 0$, $h(x) = -1$ if $x < 0$ and $h(x) \in [-1, 1]$ if $x = 0$. For the i-th element of $x^{(t+1)} - x_s$, we have

$$
\begin{aligned}
|(x^{(t+1)} - x_s)_i| &= |(I - U^{(t)}A)(x^{(t)} - x_s)_i - b^{(t)} \odot h(x_i^{(t+1)})| \\
&\leq |(I - U^{(t)}A)(x^{(t)} - x_s)_i| + |b^{(t)}|.
\end{aligned}
\tag{20}
$$

Since $\operatorname{supp}(x^{(t+1)}) \subset \mathcal{S}$, we have $\|x^{(t+1)} - x_s\|_1 = \sum_i^{|\mathcal{S}|}(x^{(t+1)} - x_s)_i|$. Thus we have

$$
\begin{aligned}
\|x^{(t+1)} - x_s\|_1 &\leq \sum_i^{\mathcal{S}}(|(I - U^{(t)}A)(x^{(t)} - x_s)_i| + |b^{(t)}|) \\
&= \sum_i^{\mathcal{S}}(|\sum_j^{S}(I - U^{(t)}A)_{ij}(x^{(t)} - x_s)_j| + |b^{(t)}|) \\
&\leq \sum_i^{\mathcal{S}}\sum_{j\neq i}^{S}|(I - U^{(t)}A)_{ij}(x^{(t)} - x_s)_j| + |S||b^{(t)}| \\
&\leq (|S| - 1)\mu(A)\|x^{(t)} - x_s\|_1 + |S|\rho^{(t)}\|U^{(t)}A(x^{(t)} - x_s)\|_1 \\
&\leq (|S| - 1)\mu(A)\|x^{(t)} - x_s\|_1 + \frac{\mu(A)|S|}{1-\mu(A)s}\|U^{(t)}A(x^{(t)} - x_s)\|_1. \\
&\leq (|S| + |S|\frac{1+\mu(A)s}{1-\mu(A)s} - 1)\mu(A)\|(x^{(t)} - x_s)\|_1.
\end{aligned}
\tag{21}
$$

The final step holds because $|S| \leq s$ and Eq.(17) hold. The $l_2$ error bound of t-th output of EBT-LISTA can be calculated as

$$
\begin{aligned}
\|x^{(t)} - x_s\|_2 &\leq \|x^{(t)} - x_s\|_1 \\
&\leq ((|S| + |S|\frac{1+\mu(A)s}{1-\mu(A)s} - 1)\mu(A))^t\|(x^{(0)} - x_s)\|_1 \\
&\leq q_0 \exp(c_1 t),
\end{aligned}
\tag{22}
$$

where $q_0 = \|x_s\|_1$ and $c_1 = \log((|S| + |S|\frac{1+\mu(A)s}{1-\mu(A)s} - 1)\mu(A))$. Compare $c_1$ with $c_0$, we have

$$\exp(c_0) - \exp(c_1) = 2\mu(A)(s - \frac{|S|}{1 - \mu(A)s}) > 0 \tag{23}$$

hold when $|S| < s(1 - \mu(A)s)$. Under this circumstance, we have

$$q_0 = \|x_s\|_1 \le |S|B < s(1 - \mu(A)s)B \le sB. \tag{24}$$

Note that $x_s$ is sampled from $\gamma(B, s)$, Eq. (23) and (24) hold with the probability with of $1 - \eta$, where

$$\eta = \frac{s - |S|}{s} \tag{25}$$
$$= \mu(A)s.$$

### A.2.2 PROOF OF LEMMA 2

For LISTA with support selection formulated as Eq.(6), there is

$$x^{(t+1)} - x_s = \text{shp}_{(b^{(t)}, p^{(t)})}((I - U^{(t)}A)x^{(t)} + U^{(t)}y) - x_s$$
$$= (I - U^{(t)}A)x^{(t)} + U^{(t)}Ax_s - x_s - b^{(t)} \odot g(x^{(t+1)}) \tag{26}$$
$$= (I - U^{(t)}A)(x^{(t)} - x_s) - b^{(t)} \odot g(x^{(t+1)}),$$

where

$$g(x) = \begin{cases} 0, i \in S_p, x_i \ne 0 \\ 1, i \notin S_p, x_i > 0 \\ -1, i \notin S_p, x_i < 0. \\ [-1, 1], x_i = 0 \end{cases} \tag{27}$$

For the i-th element of $x^{(t+1)} - x_s$, we have

$$|(x^{(t+1)} - x_s)_i| = |(I - U^{(t)}A)(x^{(t)} - x_s)_i - b^{(t)} \odot g(x_i^{(t+1)})| \tag{28}$$
$$\le |(I - U^{(t)}A)(x^{(t)} - x_s)_i| + |b^{(t)} \odot g(x_i^{(t+1)})|.$$

Since $b^{(t)} = \mu(A)\sup_{x_s} \|x^{(t)} - x_s\|_1$, same as the standard LISTA, LISTA with support selection is also "no false positive". Therefore, $\text{supp}(x^{(t+1)}) \subset S$ and $\|x^{(t+1)} - x_s\|_1 = \sum_i^S (x^{(t+1)} - x_s)_i$. Similar to Eq.(21), we have

$$\|x^{(t+1)} - x_s\|_1 \le \sum_i^S (|(I - U^{(t)}A)(x^{(t)} - x_s)_i| + |b^{(t)} \odot g(x_i^{(t+1)})|)$$
$$= \sum_i^S (|\sum_j^S (I - U^{(t)}A)_{ij}(x^{(t)} - x_s)_j| + |b^{(t)} \odot g(x_i^{(t+1)})|)$$
$$\le \sum_i^S \sum_{j \ne i}^S |(I - U^{(t)}A)_{ij}(x^{(t)} - x_s)_j| + \sum_i^S |b^{(t)} \odot g(x_i^{(t+1)})| \tag{29}$$
$$\le (|S| - 1)\mu(A)\|x^{(t)} - x_s\|_1 + \sum_i^S |b^{(t)} \odot g(x_i^{(t+1)})|.$$

From Eq.(27), we have $|g(x_i^{(t+1)})| \le 1$ and $g(x_i^{(t+1)}) = 0$ only if $i \in S_p$ and $x_i^{(t+1)} \ne 0$. We let $S_{t+1}$ denote the number of non-zero entries in $x^{(t+1)}$. Also, $P_{t+1}$ denotes the number of the largest $p^{(t+1)}\%$ elements (in absolute value) in $x^{(t+1)}$. Therefore the number of zero entries in $g(x^{(t+1)})$ is $min(S_{t+1}, P_{t+1})$. Then Eq.(29) can be calculated as

$$\|x^{(t+1)} - x_s\|_1 \le (|S| - 1)\mu(A)\|x^{(t)} - x_s\|_1 + \sum_i^S |b^{(t)} \odot g(x_i^{(t+1)})|$$
$$\le (|S| - 1)\mu(A)\|x^{(t)} - x_s\|_1 + (|S| - min(S_{t+1}, P_{t+1}))|b^{(t)}|$$
$$= (|S| - 1)\mu(A)\|x^{(t)} - x_s\|_1 + (|S| - min(S_{t+1}, P_{t+1}))\mu(A)\sup_{x_s} \|x^{(t)} - x_s\|_1. \tag{30}$$

Then, we take the supremum of Eq.(30), there is

$$\sup_{x_s} \|x^{(t+1)} - x_s\|_1 \leq (2s - 1 - min(S_{t+1}, P_{t+1}))\mu(A) \sup_{x_s} \|x^{(t)} - x_s\|_1. \tag{31}$$

Note that $\|x_s\|_1 \leq sB$ and assume $k = \arg\min_t(S_t, P_t)$, the $l_2$ upper bound of t-th output can be calculated as

$$\begin{aligned}
\|x^{(t)} - x_s\|_2 \leq \|x^{(t)} - x_s\|_1 &\leq \sup_{x_s} \|x^{(t)} - x_s\|_1 \\
&\leq \left(\prod_{i=1}^{t}(2s - 1 - min(S_i, P_i))\mu(A)\right) \sup_{x_s} \|x^{(0)} - x_s\|_1 \\
&\leq ((2s - 1 - min(S_k, P_k))\mu(A))^t sB \\
&\leq sB \exp(c_2 t),
\end{aligned} \tag{32}$$

where $c_2 = \log((2s - 1 - min(S_k, P_k))\mu(A))$. Apparently, we have $c_2 \leq c_0 = \log((2s - 1)\mu(A))$.

From Eq.(32), we have $\|x^{(t)} - x_s\|_1 \leq sB \exp(c_2 t)$, which means $l_1$ error bound can approaches to 0. Thus, there exists a $t^*$, when $t > t^*$, $\|x^{(t)} - x_s\|_1 \leq min_{i \in \mathcal{S}}(x_s)_i$. Note that $|x_i^{(t)} - (x_s)_i| \leq \|x^{(t)} - x_s\|_1$. If $i \in \mathcal{S}$, i.e., $(x_s)_i \neq 0$, there exists $x_i^{(t)} \neq 0$, which means $\mathcal{S} \subset \text{supp}(x^{(t)})$. Recall the "no false positive" property, i.e., $\text{supp}(x^{(t)}) \subset \mathcal{S}$, we can conclude that $\text{supp}(x^{(t)}) = \mathcal{S}$. Recall $P_t$ increases layerwise and $p^{(t)}$ is sufficiently large, there exists $t'$ statisfies $P_{t'} > s$, we let $t_0 = \max(t^*, t')$, if $t > t_0$, there exists $P_t \geq |\mathcal{S}|$ and $\text{supp}(x^{(t)}) = \mathcal{S}$. Under this circumstance, if $i \in \mathcal{S}$, we have $x_i^{(t)} \neq 0$ and $i \in S_{p_t}$, which means every element in $\mathcal{S}$ will be selected as support. Therefore, we have

$$\begin{aligned}
x_i^{(t+1)} - (x_s)_i &= \text{shp}_{(b^{(t)}, p^{(t)})}(((I - U^{(t)}A)x^{(t)} + U^{(t)}Ax_s)_i) - (x_s)_i \\
&= ((I - U^{(t)}A)x^{(t)} + U^{(t)}Ax_s)_i - (x_s)_i \\
&= ((I - U^{(t)}A)(x^{(t)} - x_s))_i.
\end{aligned} \tag{33}$$

We let $x_\mathcal{S} \in \mathbb{R}^{|\mathcal{S}|}$ denote the vector that keeps the elements with indices of $x$ in $\mathcal{S}$ and remove the others. Similarly, we let $M(\mathcal{S}, \mathcal{S}) \in \mathbb{R}^{|\mathcal{S}| \times |\mathcal{S}|}$ denote the submatrix of matrix $M$ which keeps the row and column if the index belongs to $\mathcal{S}$. Then, we have

$$\begin{aligned}
\|x^{(t+1)} - x_s\|_2 &= \|(x^{(t+1)} - x_s)_\mathcal{S}\|_2 \\
&= \|((I - U^{(t)}A)(x^{(t)} - x_s))_\mathcal{S}\|_2 \\
&= \|(I - U^{(t)}A)(\mathcal{S}, \mathcal{S})(x^{(t)} - x_s)_\mathcal{S}\|_2 \\
&\leq \|(I - U^{(t)}A)(\mathcal{S}, \mathcal{S})\|_2 \|(x^{(t)} - x_s)_\mathcal{S}\|_2 \\
&= C\|(x^{(t)} - x_s)\|_2,
\end{aligned} \tag{34}$$

where $C = \|(I - U^{(t)}A)(\mathcal{S}, \mathcal{S})\|_2$. Further we have $C \leq \|(I - U^{(t)}A)(\mathcal{S}, \mathcal{S})\|_F \leq \sqrt{|\mathcal{S}|^2\mu(A)^2} \leq s\mu(A)$.

### A.2.3 Proof of Theorem 2

For the EBT-LISTA with support selection formulated in Eq. (11), similar to Eq. (26) and (28), we have

$$\begin{aligned}
|(x^{(t+1)} - x_s)_i| &= |(I - U^{(t)}A)(x^{(t)} - x_s)_i - b^{(t)} \odot g(x_i^{(t+1)})| \\
&\leq |(I - U^{(t)}A)(x^{(t)} - x_s)_i| + |b^{(t)} \odot g(x_i^{(t+1)})|,
\end{aligned} \tag{35}$$

where $b^{(t)} = \rho^{(t)}\|U^{(t)}(Ax^{(t)} - y)\|_1 = \frac{\mu(A)}{1 - \mu(A)s}\|U^{(t)}A(x^{(t)} - x_s)\|_1$. Same as the origin EBT-LISTA, EBT-LISTA with support selection is also "no false positive" and Eq. (17) hold either. Therefore $\|U^{(t)}A(x^{(t)} - x_s)\|_1 \leq (1 + |S|\mu(A))\|x^{(t)} - x_s\|_1 \leq (1 + s\mu(A))\|x^{(t)} - x_s\|_1$. Similar to

Eq. (29) and (30), there is

$$
\begin{aligned}
\|x^{(t+1)} - x_s\|_1 &= \sum_i^{\mathcal{S}} |(x^{(t+1)} - x_s)_i| \\
&\leq \sum_i^{\mathcal{S}} (|(I - U^{(t)}A)(x^{(t)} - x_s)_i| + |b^{(t)} \odot g(x_i^{(t+1)})|) \\
&= \sum_i^{\mathcal{S}} (|\sum_j^{S} (I - U^{(t)}A)_{ij}(x^{(t)} - x_s)_j| + |b^{(t)} \odot g(x_i^{(t+1)})|) \\
&\leq \sum_i^{\mathcal{S}} \sum_{j \neq i}^{S} |(I - U^{(t)}A)_{ij}(x^{(t)} - x_s)_j| + \sum_i^{\mathcal{S}} |b^{(t)} \odot g(x_i^{(t+1)})| \\
&\leq (|\mathcal{S}| - 1)\mu(A)\|x^{(t)} - x_s\|_1 + (|\mathcal{S}| - min(S_{t+1}, P_{t+1}))|b^{(t)}| \\
&\leq (|\mathcal{S}| - 1)\mu(A)\|x^{(t)} - x_s\|_1 + (|\mathcal{S}| - min(S_{t+1}, P_{t+1}))\mu(A)\frac{1 + \mu(A)s}{1 - \mu(A)s}\|(x^{(t)} - x_s)\|_1 \\
&\leq (\frac{2}{1 - \mu(A)s}|\mathcal{S}| - \frac{1 + \mu(A)s}{1 - \mu(A)s}min(S_{t+1}, P_{t+1}) - 1)\mu(A)\|(x^{(t)} - x_s)\|_1.
\end{aligned}
\tag{36}
$$

Similar to Eq. (32), the $l_2$ error bound can be calculated as

$$
\begin{aligned}
\|x^{(t)} - x_s\|_2 &\leq \|x^{(t)} - x_s\|_1 \\
&\leq \prod_{i=1}^{t} \left[ (\frac{2}{1 - \mu(A)s}|\mathcal{S}| - \frac{1 + \mu(A)s}{1 - \mu(A)s}min(S_i, P_i) - 1)\mu(A) \right] \|(x^{(0)} - x_s)\|_1 \\
&\leq \left[ (\frac{2}{1 - \mu(A)s}|\mathcal{S}| - \frac{1 + \mu(A)s}{1 - \mu(A)s}min(S_k, P_k) - 1)\mu(A) \right]^t \|x_s\|_1 \\
&\leq q_1 \exp(c_3 t),
\end{aligned}
\tag{37}
$$

where $q_1 = \|x_s\|_1$ and $c_3 = \log((\frac{2}{1 - \mu(A)s}|\mathcal{S}| - \frac{1 + \mu(A)s}{1 - \mu(A)s}min(S_k, P_k) - 1)\mu(A))$.

Compare $c_3$ with $c_2$, we have

$$
\begin{aligned}
\exp(c_2) - \exp(c_3) &= 2\mu(A)(s - \frac{|S|}{1 - \mu(A)s} + \frac{2\mu(A)s}{1 - \mu(A)s}min(S_k, P_k)) \\
&\geq 2\mu(A)(s - \frac{|S|}{1 - \mu(A)s}) > 0
\end{aligned}
\tag{38}
$$

hold when $|S| < s(1 - \mu(A)s)$. Under this circumstance, we have

$$
q_1 = \|x_s\|_1 \leq |S|B < s(1 - \mu(A)s)B \leq sB.
\tag{39}
$$

Note that $x_s$ is sampled from $\gamma(B, s)$, Eq. (38) and (39) hold with the probability with of $1 - \eta$, where

$$
\begin{aligned}
\eta &= \frac{s - |S|}{s} \\
&= \mu(A)s.
\end{aligned}
\tag{40}
$$

Similar to LISTA with support selection, there exists a $t^{**}$, when $t > t^{**}$, $\|x^{(t)} - x_s\|_1 \leq min_{i \in \mathcal{S}}(x_s)_i$. Therefore, $supp(x^{(t)}) = \mathcal{S}$. Recall that $c_3 < c_2$ holds with the probability of $1 - \eta$, we have $t^{**} < t^*$ hold with the probability of $1 - \eta$. When we use the same settings for $p^{(t)}$ as LISTA-SS, we have same $t'$ satisfying $P_{t'} > s$. Let $t_1 = max(t^{**}, t')$, we have $t_1 \leq t_0$ with the probability of $1 - \eta$. When $t > t_1$, we have $x_i^{(t)} \neq 0$ and $i \in S_{p_t}$. Same as Eq. (33) and (34), there is

$$
\|x^{(t+1)} - x_s\|_2 \leq C\|(x^{(t)} - x_s)\|_2,
\tag{41}
$$

where $C = \|(I - U^{(t)}A)(\mathcal{S}, \mathcal{S})\|_2 \leq \|(I - U^{(t)}A)(\mathcal{S}, \mathcal{S})\|_F \leq \sqrt{|\mathcal{S}|^2\mu(A)^2} \leq s\mu(A)$.

